# Mushrooms Do Produce Flavonoids: Metabolite Profiling and Transcriptome Analysis of Flavonoid Synthesis in the Medicinal Mushroom *Sanghuangporus baumii*

**DOI:** 10.3390/jof8060582

**Published:** 2022-05-29

**Authors:** Shixin Wang, Zengcai Liu, Xutong Wang, Ruipeng Liu, Li Zou

**Affiliations:** Department of Forest Conservation, College of Forestry, Northeast Forestry University, No. 26 Hexing Road, Harbin 150040, China; 2017010037@nefu.edu.cn (S.W.); 1758458181@nefu.edu.cn (Z.L.); woshixiannv1234@sina.com (X.W.); liuruipeng@nefu.edu.cn (R.L.)

**Keywords:** flavonoids, mushroom, metabolic engineering, *Sanghuangporus baumii*, phenylalanine ammonia lyase

## Abstract

Mushrooms produce a large number of medicinal bioactive metabolites with antioxidant, anticancer, antiaging, and other biological activities. However, whether they produce flavonoids and, if so, how they synthesize them remains a matter of some debate. In the present study, we combined flavonoid-targeted metabolomics and transcriptome analysis to explore the flavonoid synthesis in the medicinal mushroom *Sanghuangporus baumii*. The *S. baumii* synthesized 81 flavonoids on a chemically defined medium. The multiple classes of flavonoids present were consistent with the biosynthetic routes in plants. However, paradoxically, most of the genes that encode enzymes involved in the flavonoid biosynthetic pathway are missing from *S. baumii*. Only four genes related to flavonoid synthesis were found in *S. baumii*, among which phenylalanine ammonia lyase gene (*PAL*) is a key gene regulating flavonoid synthesis, and overexpression of *SbPAL* increases the accumulation of flavonoids. These results suggest that the flavonoid synthesis pathway in *S. baumii* is different from that in known plants, and the missing genes may be replaced by genes from the same superfamilies but are only distantly related. Thus, this study provides a novel method to produce flavonoids by metabolic engineering using mushrooms.

## 1. Introduction

Flavonoids are secondary metabolites produced by plants and have antioxidant, anti-inflammatory [1], and anticancer [2] activities. It is widely believed that mushrooms also contain flavonoids. However, in 2016, Gil-Ramirez et al. tested 24 mushroom species from 19 genera using HPLC–MS, and no flavonoids were identified [3]. In addition, they reported that the coding sequence of chalcone synthase (CHS), a key enzyme in the flavonoid biosynthetic pathway, is not found in any mushroom genome database. Therefore, the authors concluded that mushrooms cannot synthesize flavonoids. Conversely, in a genome-wide survey of the fungal kingdom reported in 2020, Tapan identified all the gene sequences associated with flavonoid biosynthesis [4]. However, most of the sequences the author presented were derived from filamentous fungi, which do not form mushrooms. Accordingly, whether mushrooms can synthesize flavonoids remains unknown.

In plants, flavonoids are converted from phenylalanine through the phenylpropanoid pathway. First, phenylalanine is deaminated under the action of phenylalanine ammonia-lyase (*PAL*) to produce cinnamic acid. Cinnamic acid forms 4-coumaroyl-CoA under the action of cinnamate-4-hydroxylase (C4H) and 4-coumarate-CoA ligase (4CL). Then, CHS catalyzes the condensation of three malonyl-CoA units with 4-coumaroyl-CoA to yield chalcone, which is catalyzed by chalcone isomerase (CHI) to form flavanone [5]. Finally, flavanone forms a series of flavonoids under the catalysis of different enzymes. All compounds with the basic structural skeleton of flavonoids share this common biosynthetic pathway. However, the genes that encode most of the enzymes involved in the flavonoid synthesis pathway are missing from the genome of mushrooms. While *PAL*-like proteins have been found in a few species of mushrooms, such as *Agaricus bisporus* and *Pleurotus eryngii*, their catalytic functions are unknown [3]. Therefore, it is not clear how flavonoids are synthesized in mushrooms.

*Sanghuangporus baumii* is a medicinal mushroom that parasitizes the trunk of *Syringa reticulata* and forms yellow fruiting bodies (Figure 1A). In recent decades, *S. baumii* has received widespread attention for its anti-cancer and anti-inflammatory effects [6,7], and it is believed that flavonoids are the main bioactive substances responsible for these activities. Previous studies have focused on the fermentation production of total flavonoids from *S. baumii*, but the specific flavonoids that can be synthesized from *S. baumii* and how they are synthesized remain unknown. In addition, it is important to note that flavonoids may be present in *S. baumii* by two possible actions; biosynthesis within *S. baumii* or absorption from the host plant/culture medium. Therefore, in order to determine the classes of flavonoids synthesized by *S. baumii*, it is necessary to analyze *S. baumii* grown on a chemically defined medium. 

Recent advances in omics technologies have enabled plant biologists to discover many important plant-based medicinal metabolites [8] and elucidate their complete biosynthetic pathways [9]. Furthermore, our previous studies have shown that aeration is an effective method to promote the accumulation of flavonoids in the mycelia of *S. baumii*. Thus, metabolite profiling and transcriptome analysis of *S. baumii* mycelia after different aeration times are used in this study to investigate flavonoid biosynthesis in *S. baumii*. 

The aim of this study was to explore the flavonoid synthesis pathway in *S. baumii* and provide a basis for its utilization of this mushroom in the production of flavonoids. Accordingly, we have demonstrated that *S. baumii* can synthesize 81 flavonoids and identified four genes related to flavonoid synthesis. The function of the key gene *PAL* was verified, and the overexpression of *PAL* was found to promote the accumulation of flavonoids in the mycelia of *S. baumii*.

## 2. Materials and Methods

### 2.1. Strain and Culture Conditions

The *S. baumii* strain was identified by morphological and molecular identification (GenBank accession number KP974834) and preserved at the College of Forestry, Northeast Forestry University, Harbin, China. The strain was inoculated to culture plates and cultured at 28 °C under dark conditions. Chemically defined medium (semisolid): dextrose 20 g, soluble starch 10 g, KNO_3_ 1 g, K_2_HPO_4_ 1 g, glycine 0.5 g, NaCl 0.5 g, MgSO_4_ 0.5 g, FeSO_4_ 0.01 g, agar 6 g, H_2_O 1 L. PDA medium (semisolid): potato infusion 20 g, dextrose 20 g, agar 6 g, H_2_O 1 L. 

### 2.2. Aeration Treatment

In biological laboratories, culture plates are often wrapped with sealing film to avoid dehydration and contamination [10]. *S. baumii* forms white colonies on culture plates wrapped by sealing film. The process of removing the sealing film and exposing the mycelia to the outside air is called aeration. Inoculated culture plates were wrapped with sealing film (LINGS company, Hangzhou, China) and cultured for 12 d before the mycelia was harvested. On the 9th, 10th, and 11th day of culturing, the sealing film of some plates was removed so that the plates were aerated for 72, 48, and 24 h, respectively. The aeration time of the plates without the sealing film removed during culturing was defined as 0 h. The mycelia of chosen colonies was harvested by scraping from the semisolid medium.

An oxygen detector AR8100 (SMART SENSOR, Dongguan, China) was used to measure the O_2_ contents of aerated and unaerated culture plates after 12 d of cultivation. Capillary gas chromatography (Agilent Technologies, Santa Clara, CA, USA) was used to measure CO_2_ content under the conditions described by Chai [11].

### 2.3. Determination of Bioactive Substance Contents and Enzyme Activity

The mycelia used for the determination of bioactive substances was dried to a constant weight at 50 °C, ground into powder, and sifted through a 60-mesh sieve. The total flavonoid value was determined by the aluminum nitrate colorimetric method [12]. The total triterpenoid value was determined by the vanillin–acetic acid method described elsewhere [13].

The mycelia used for the determination of enzyme activity was ground with a homogenizer in an ice bath. *PAL* activity was determined by measuring the production rate of *trans*-cinnamic acid. A *PAL* activity unit is defined as the amount of enzyme required for a 0.01 change in absorbance at 290 nm for 60 min. Polyphenol oxidase (PPO) activity was determined by measuring the oxidation rate of catechol [14]. A PPO activity unit is defined as the amount of enzyme required for a 0.01 change in absorbance at 525 nm for 1 min. Catalase (CAT) activity was determined by measuring the decomposition rate of H_2_O_2_ [15]. A CAT activity unit (U) is defined as the amount of enzyme required for a 0.01 change in absorbance at 240 nm for 1 min.

### 2.4. Analysis of Flavonoids by UPLC–MS/MS

UPLC–MS/MS (SHIMADZU Nexera X2; Tandem mass spectrometry, Applied Biosystems 4500 QTRAP) was used for qualitative and relative quantitative analysis of the flavonoids in *S. baumii*. The freeze-dried mycelia was crushed into powder and extracted overnight at 4 °C with methanol/water (70%, *v*/*v*). UPLC–MS/MS analysis was performed by Metware Biotechnology Co., Ltd. (Wuhan, China). The multiple reaction monitoring (MRM) method was previously described by Zhao [16]. Flavonoids were identified using the self-built database (MetWare Biotechnology Co.) through parameters such as *m*/*z* value, retention time, and fragmentation mode. The filtering conditions for differential accumulated metabolites were as follows: absolute log 2 (fold change) ≥ 1, *p*-value < 0.05, and variable importance in projection (VIP) ≥ 1.

### 2.5. RNA Sequencing and Quantitative Real-Time Polymerase Chain Reaction (qRT-PCR) Analysis

Total RNA was isolated using the RNA prep Pure Plant Plus Kit (TIANGEN, Beijing, China), and the concentration and integrity of total RNA were determined by 1% agarose gel electrophoresis and nano-photometry (IMPLEN, Westlake Village, CA, USA). The cDNA library construction, sequencing, and annotation was performed by Metware Biotechnology Co., Ltd., as described in detail by Su [17]. An absolute log2 (fold change) ≥1 and false discovery rate (FDR) < 0.05 were used as thresholds for the identification of differentially expressed genes. Pathway analysis was performed to elucidate the significant pathways of differentially expressed genes using the Kyoto Encyclopedia of Gene and Genomes (KEGG) (http://www.genome.jp/kegg accessed on 16 January 2021). 

The primers used for qRT-PCR to verify gene expression are shown in Appendix A, and the *α*-tubulin gene was used as an internal reference. First-strand cDNA was synthesized using PrimeScript ™ RT reagent Kit (TaKaRa, Beijing, China). The relative expression of genes was calculated using the 2^−ΔΔCT^ method. 

### 2.6. PAL Gene Cloning and Bioinformatics Analysis

cDNA was synthesized with PrimeScript II first-strand cDNA synthesis kit (TaKaRa). Primers *PAL*-F and *PAL*-R were designed to obtain the complete sequence (Appendix A). Theoretical molecular weight, isoelectric point, and absorption coefficient were predicted using ExPASy Proteomics Server (http://www.expasy.ch/tools/protparam.html accessed on 5 March 2021). The 3D structure was predicted using SWISS-MODEL online homologous modelling software (http://swissmodel.expasy.org accessed on 5 March 2021). Homologous sequences were downloaded to construct a neighbor-joining tree with MEGA 5.0 software (Version number 5.0, Mega Limited, Auckland, New Zealand).

### 2.7. Purification and Functional Validation of PAL Protein

The *PAL* open reading frame was incorporated into the pET-32a vector (TaKaRa, Beijing, China) and transferred into *Escherichia coli* (TaKaRa, Beijing, China) to express the recombinant protein (*PAL*-B, *PAL*-H, Appendix A). Protein expression was triggered by the addition of isopropyl-β-D-thiogalactoside (IPTG, TIANGEN, Beijing, China) to an ultimate 1 mmol/L concentration and keeping the temperature at 18 °C for 10 h. The Capturem His-Tagged Purification Miniprep Kit (TaKaRa, Beijing, China)was used to purify the *PAL* protein. The results were observed by 12% sodium dodecyl sulphate polyacrylamide gel electrophoresis (SDS–PAGE). The reaction conditions for verifying the catalytic function of *PAL* were as follows: 2 mL 0.1 mol/L borate buffer (pH 8.8); 1 mL 0.02 mol/L *L*-phenylalanine (pre-dissolved in borate buffer); 100 μL purified protein (100 μL borate buffer was used as a blank control); water bath at 30 °C for 30 min. The reaction liquid was analyzed for the formation of *trans*-cinnamic acid using liquid chromatography (Waters 1525) with a C18 column (150 × 4.6 mm, 5 μm).

### 2.8. Plasmid Construction and Transformation of S. baumii Protoplasts

The plasmid pAN7-1 contains the *E. coli* hygromycin B phosphotransferase gene (*hph*) fused to the promoter of *Aspergillus nidulans* glyceralde-hyde-3-phosphate dehydrogenase gene (*gpd*). Expression of *hph* confers resistance to the hygromycin B. The plasmid pAN7-*PAL* was constructed based on pAN7-1. Primers were designed to linearize pAN7-1 by PCR and remove the *hph* sequence (pAN7-F, pAN7-R, Appendix A). The linearized plasmid was then linked to the open reading frame of *PAL* using an In-Fusion^®^ HD Cloning Kit (TaKaRa) (*PAL*-iF-F, *PAL*-iF-R, Appendix A). The two plasmids were co-transformed into protoplasts of *S. baumii* using polyethylene glycol (PEG)-mediated transformation using the method described by Li [18]. The transformed protoplasts were cultured on MYG medium plates containing 3 mg∙L^−1^ hygromycin B at 28 °C for 5–7 d to regenerate positive single colonies. Positive single colonies were sub-cultured individually onto PDA plates containing 3 mg∙L^−1^ hygromycin B to obtain resistant strains. PCR amplification of the resistant strains was performed to confirm the successful transformation of pAN7-1 and pAN7-*PAL*. MYG medium: 1% maltose, 0.4% glucose, 0.4% yeast extract, and 0.6 mol/L mannitol.

### 2.9. Statistical Analysis

Quantitative data are presented as mean values from three independent experiments with standard deviation of the means. Statistical analysis was performed using Tukey tests. Values were considered statistically significantly different at the *p* < 0.05 level.

## 3. Results

### 3.1. Determination of Flavonoids in S. baumii by UPLC–MS/MS

To verify whether *S. baumii* can synthesize flavonoids, we determined the flavonoid components of mycelia grown on the chemical defined medium by UPLC–MS/MS. After 23 d culturing, *S. baumii* formed yellow colonies on the chemical defined medium (Figure 1B). Eighty-one flavonoids were identified in *S. baumii* by mass spectrometry, including 25 flavones, 23 flavonols, 10 flavanols, eight isoflavones, five flavanones, four flavanonols, two chalcones, three proanthocyanidins, and one isoflavanone (Appendix A). Figure 2 shows the MRM detection of multimodal maps. Since the chemically defined medium did not contain any flavonoids, the detected flavonoids were synthesized by *S. baumii*. Thus, these results demonstrate that *S. baumii* can synthesize flavonoids.

### 3.2. Aeration Promotes Flavonoid Accumulation in S. baumii

Next, we determined the levels of total flavonoids and other biochemical indexes in the mycelia of *S. baumii* after different aeration times. As shown in Figure 3, aeration increases the O_2_ concentration and decreases the CO_2_ concentration in the plate (before aeration: O_2_ 11.27 ± 1.59%, CO_2_ 2.23 ± 0.33%; after aeration: O_2_ 20.35 ± 0.05%, CO_2_ 0.07 ± 0.02%). With increasing aeration time, the colour of the *S. baumii* colonies gradually changes from white to dark-yellow (Figure 3A), and the total flavonoid content and *PAL* activity (a key enzyme in flavonoid synthesis) increase significantly, suggesting that aeration induces flavonoid synthesis (Figure 3B). However, the content of total triterpenoids, another secondary metabolite, does not change significantly under aeration. The biomass, PPO activity, and CAT activity increase with aeration time (Figure 3B). PPO catalyzes the oxidation of several phenols to *o*-quinones and plays a role in the defence response to biotic and abiotic stresses. CAT catalyzes the decomposition of hydrogen peroxide, which is an essential signal-transduction molecule and increases during stress. The increased activity of PPO and CAT indicate that *S. baumii* is stressed by aeration and may increase its stress resistance by accumulating specific secondary metabolites, such as flavonoids. These results indicates that aeration is an effective method to induce flavonoid synthesis in *S. baumii*.

### 3.3. Metabolite Profiling and Transcriptome Analysis Reveals the Flavonoid Synthesis Pathway in S. baumii 

To clarify the flavonoid synthesis pathway in *S. baumii*, metabolomics and transcriptomics were employed to analyze the differences in flavonoid composition and gene expression between mycelia aerated for 0 or 48 h. Sixteen metabolites with significant content differences were identified by metabonomics analysis, including three flavonols, three flavones, two flavanones, one chalcones, one isoflavones, one flavanonols, and five tannins (Figure 4A). The contents of nine flavonoids are significantly increased in the mycelia aerated for 48 h, among which the contents of 3-*O*-acetylpinobanksin, kaempferol-3-*O*-(6′′-acetyl)-glucoside, and genistein-7-*O*-galactoside increase 48,333-, 14,233-, and 12,488-fold, respectively. These results are consistent with our previous conclusion that aeration promotes flavonoid accumulation.

A transcriptomic comparison of the mycelia aerated for 0 and 48 h was conducted. A total of 47,046 genes were identified after assembly and annotation, including 6435 differentially expressed genes (2552 upregulated and 3883 downregulated; Appendix A). KEGG pathway enrichment analysis showed that these differentially expressed genes are mainly enriched in metabolic pathways and biosynthesis of secondary metabolites (Red, italic letters in Figure 4B). 

These results indicate that aeration induces the synthesis of secondary metabolites in *S. baumii*. However, of the genes in the flavonoid synthesis pathway (Figure 4C), only *PAL*, *4CL*, *CHI*, and isoflavone reductase (*IFR*) are identified in *S. baumii*. Thus, most of the genes involved in flavonoid synthesis in plants, such as *CHS*, are not found in the transcriptome or, in fact, in the genome of *S. baumii* (GenBank: GCA_001481415.2). 

Four differentially expressed genes (three genes in the flavonoid synthesis pathway and the polyketone synthase gene (*PKS*)) were selected to validate the transcriptomic results, and the selected genes show a high correlation between their qRT-PCR and transcriptomic data sets (Figure 4D).

Combined metabolomics and transcriptome analysis revealed that the multiple classes of flavonoids in *S. baumii* are consistent with the presence of biosynthetic routes that are commonly found in plants, but most of the genes involved in plant flavonoid biosynthesis are missing. Accordingly, we cloned and analyzed three of the differentially expressed genes, *PAL*, *4CL*, and *IFR* (data for *4CL* and *IFR* not shown in this study), and further investigated the function of *SbPAL*.

### 3.4. Functional Analysis of SbPAL

We investigated the function of *SbPAL* using bioinformatics and experimental methods. The *SbPAL* fragment was amplified, and an open reading frame of 2493 bp encoding an 830-amino-acid polypeptide was predicted (Appendix A). The predicted molecular weight of the protein is 87.75 kDa for the molecular formula C_3837_H_6160_N_1070_O_1225_S_27_. The theoretical isoelectric point of the protein is 5.63, and the instability index is 37.77, classifying the protein as stable. Based upon sequence homology tools (http://www.ncbi.nlm.nih.gov/BLAST, accessed on 11 February 2021), the *SbPAL* amino-acid sequence shows a high identity with PALs in fungi in the GeneBank database and shares 75.67% identity with *PAL* of *Inonotus obliquus* (AJD20419.1). 

A phylogenetic tree (Figure 5A) was constructed based on the *PAL* sequences of 16 species (Appendix A), of which 11 species belong to Basidiomycota, four species belong to Ascomycota, and *Arabidopsis thaliana* is an outgroup plant (The *PAL* of *S. baumii* is shown in red letters in Figure 5A). The phylogenetic results indicate that *PAL*-sequence similarity correlates well with species. 

The 3D structure of *Sb**PAL* was predicted using the crystal structure of *Rhododporidium toruloides PAL* (1y2m.1.A) as a template. Like those for other *PAL* families, *Sb**PAL* is a homo-tetrameric protein mainly comprising alpha helices. The monomer and tetramer structure of *Sb**PAL* are shown in Figure 5B.

The recombinant protein of *Sb**PAL* was successfully expressed in *E. coli* BL21 using pET-32a as a vector. After purification, a clear protein band for the expected molecular weight (87.75 kDa + tag protein 22.68 kDa) was observed on the 12% SDS-PAGE gel (Figure 5C). *PAL* catalyzes the deamination of *L*-phenylalanine to form *trans*-cinnamic acid. The product *trans*-cinnamic acid is present in the reaction solution containing the purified protein while the blank control shows no such peak (Figure 5D). These results demonstrate that the *Sb**PAL* has the catalytic function expected for *PAL*.

### 3.5. Overexpression of the SbPAL Promotes Flavonoid Accumulation in S. baumii

The effect of the overexpression of *SbPAL* on flavonoid synthesis in *S. baumii* was studied. The transformed protoplasts regenerated positive single colonies on MYG plates containing hygromycin B, and then the single colonies were sub-cultured on PDA plates containing hygromycin B for 30 days to obtain hygromycin-B-resistant strains (Figure 6A). PCR amplification was used to detect the fragment of the *hph* gene and the *SbPAL* gene driven by a promoter of the *gpd* gene to verify whether the resistant strain was successfully transgenic (Appendix A). Two *SbPAL*-overexpressing (OP) strains were obtained by this process. As shown in Figure 6, there are no significant differences in biomass and macroscopical characteristics between the wild-type (WT) and OP strains (Figure 6B,C). The total flavonoid content, *PAL* enzyme activity, and *PAL* expression level of the OP strain are 1.39-, 1.49-, and 2.38-fold higher than those of the WT strain, respectively, indicating that overexpression of *PAL* increases the transcription level of the *PAL* gene and *PAL* enzyme activity, ultimately increasing the flavonoid content of *S. baumii* (Figure 6C). In addition, the expression levels of *4CL* and *PKS* in the OP strain increase, which is consistent with the results shown in Figure 4D. These results suggest that overexpression of *SbPAL* promotes flavonoid biosynthesis and accumulation in *S. baumii*.

## 4. Discussion

Flavonoids are the most diverse common secondary metabolites in plants. They provide protection against UV radiation and were originally thought to only exist in land plants [19]. However, flavonoid biosynthesis is not unique to land plants, and it has gradually emerged that some aquatic algae and fungi can also synthesize flavonoids [20]. Mushrooms produce a large number of secondary metabolites, but whether they produce flavonoids has remained a matter of debate. However, this study has proved that the medicinal mushroom *S. baumii* can synthesize flavonoids and provided details about the synthesis pathway used.

Eighty-one flavonoids were detected in the mycelia of *S. baumii* cultured on chemically defined medium, demonstrating that *S. baumii* can synthesize flavonoids. The flavonoids detected include glycosylated or acylated genistein [21], pinobanksin [22], and kaempferol [23], which are well studied anti-cancer agents. The presence of these flavonoids provides a rational explanation for the medicinal effects of *S. baumii*. In a recent survey of 24 mushroom species, no flavonoids were detected, but that does not prove that all mushrooms do not contain flavonoids. Fungi show high levels of diversity in species, tropisms, and metabolites [24]. Furthermore, it is estimated that there are ~140,000 species of macrofungi in nature, but only ~20,000 species have been described so far, with far fewer being subjected to metabolic analysis [25]. Thus, with ongoing research, more flavonoid-producing mushrooms may be found in nature. 

In plants, flavonoids provide pigmentation for organs and participate in response to biotic and abiotic stresses [26]. The flavonoids in *S. baumii* may have similar functions. In the experiment demonstrating that aeration promotes flavonoid accumulation in *S. baumii*, an increase in flavonoid content was accompanied by a gradual yellowing of colony color (Figure 3A). We speculate that flavonoids give *S. baumii* its yellow color, but it is unclear which specific flavonoids are responsible. 

All known flavonoid compounds, of which there are more than 7000, have the same basic structural skeleton because they share a common biosynthetic pathway, as shown in Figure 4C. The multiple classes of flavonoids synthesized by *S. baumii* are consistent with the flavonoids in this pathway. However, *S. baumii* lacks most of the enzymes involved in this flavonoid synthesis pathway. A similar paradox exists for freshwater algal *Penium margaritaceum* [27]. The flavonoid compounds found in *P. margaritaceum* are thought to be related to the evolutionary trajectory of plants toward terrestrialization. However, a number of the genes involved in flavonoid synthesis are absent from the genome of *P. margaritaceum*. These missing genes are likely to be replaced by distantly-related genes from the same superfamilies. For example, although the *CHS* gene is not found in the transcriptome of *S. baumii*, the *PKS* gene, which is from the same family as the *CHS* gene, is found. PKS catalyzes the condensation of acyl-coA to produce polyketides, and the catalytic mechanism is similar to that of CHS. In future work, we intend to determine whether *S. baumii* PKS exhibits CHS-like catalytic functionality.

Overexpression of key genes in the synthesis pathway has been used as an effective genetic tool to elevate the levels of various metabolites [28]. In plants, *PAL* is a rate-limiting enzyme of flavonoid biosynthesis, and an increase in *PAL* expression can cause the accumulation of flavonoids [29,30,31]. Similar results were obtained for our study on the overexpression of *SbPAL* in *S. baumii*. Compared with the WT strain, the *PAL* expression level and *PAL* enzyme activity in the OP strain are increased, ultimately leading to an increase in total flavonoid content. These results indicate that *SbPAL* is a key point for flux into flavonoid biosynthesis in terms of the genetic control of secondary metabolism in *S. baumii*.

In conclusion, *S. baumii* can synthesize flavonoids, and, to our knowledge, this is the first study to demonstrate that mushrooms can synthesize flavonoids. The flavonoid synthesis pathway in *S. baumii* is different from that in known plants, and many enzymes that catalyze the pathway in plants are absent from *S. baumii*. This deserves further study. *PAL* is a key gene in the flavonoid synthesis pathway of *S. baumii*, and overexpression of *PAL* increases the accumulation of flavonoids. Overall, this study indicates that *S. baumii* (and, potentially, other mushrooms) shows promise as a microbial cell factory in the production of flavonoids.

## Figures and Tables

**Figure 1 jof-08-00582-f001:**
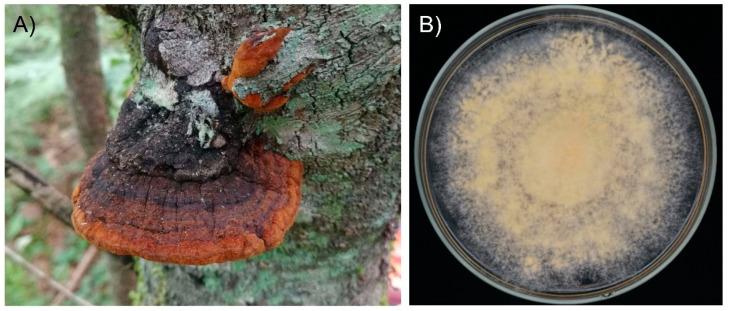
The fruiting body and colony of *Sanghuangporus baumii*. (**A**), The fruiting body of *S. baumii*. (**B**), Colony of *S. baumii* on chemically defined medium.

**Figure 2 jof-08-00582-f002:**
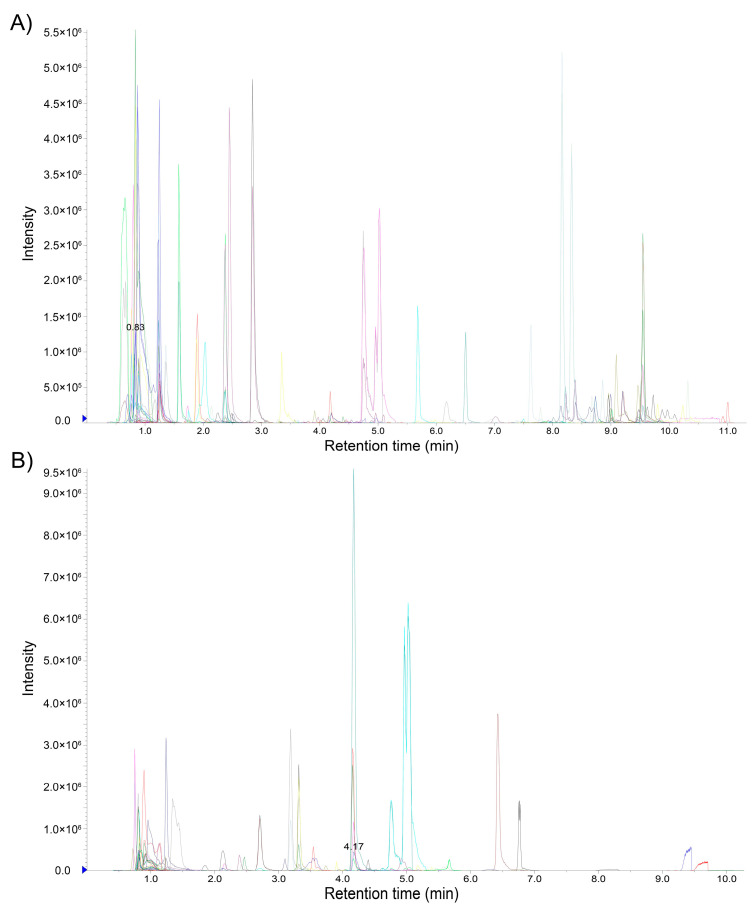
MRM detection of multimodal maps. (**A**), Cationic mode. (**B**), Anion mode.

**Figure 3 jof-08-00582-f003:**
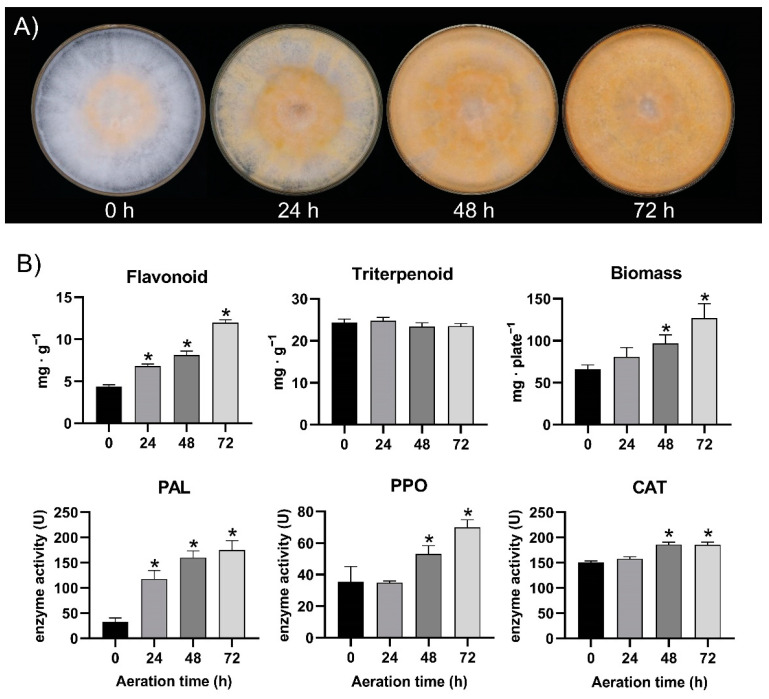
Effects of aeration time on the appearance and biochemical parameters of *S. baumii* mycelia. (**A**) Effect of aeration time on the appearance of a *S. baumii* colony. (**B**) Effects of aeration time on the biochemical parameters of *S. baumii* mycelia. *PAL*: phenylalanine ammonia lyase; PPO: polyphenol oxidase; CAT: catalase. The values with * are significantly different (*p* < 0.05).

**Figure 4 jof-08-00582-f004:**
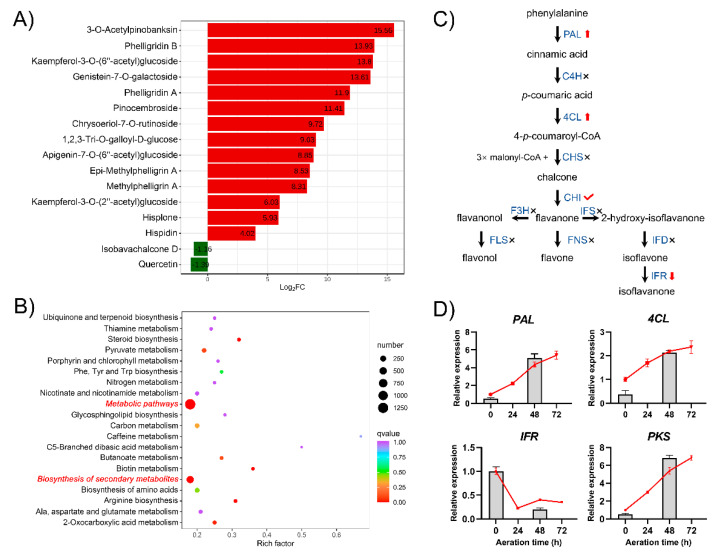
Metabolomics and transcriptome analysis of *S. baumii* mycelia aerated for 0 and 48 h. (**A**) Sixteen differential metabolites in the mycelia aerated for 0 h and 48 h. (**B**) Top 20 KEGG pathways with the most significant enrichment for mycelia aerated for 0 and 48 h. (**C**) Flavonoid biosynthesis pathway in *S. baumii*. Enzymes in the pathway are shown in blue. The red arrows up or down indicate increased or decreased gene expression after aeration, the red check mark indicates that the gene expression level has no significant change, and the black crosses indicate absence of such a homolog in *S. baumii*. *PAL*, phenylalanine ammonia lyase; C4H, cinnamate 4-hydroxylase; 4CL, 4-coumarate-CoA ligase; CHS, chalcone synthase; CHI, chalcone isomerase; FNS, flavone synthase; F3H, flavanone-3-hydroxylase; FLS, flavonol synthase; IFS, 2-hydroxy isoflavone synthase; IFD, 2-hydroxyl isoflavone dehydrase; IFR, isoflavone reductase. (**D**) Verification of differentially expressed genes by quantitative real-time polymerase chain reaction (qRT-PCR). Bars show transcriptome data and lines show qRT-PCR data. *PKS*, polyketone synthase gene.

**Figure 5 jof-08-00582-f005:**
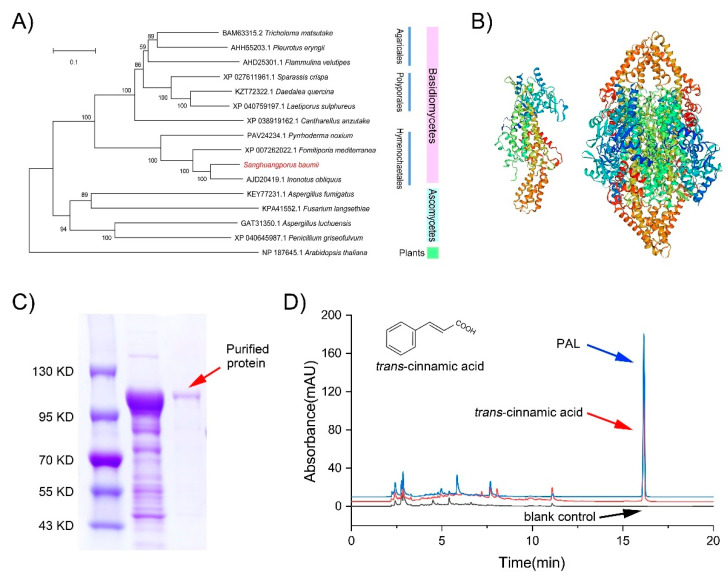
Functional analysis of *SbPAL*. (**A**) Phylogenetic tree constructed based on the *PAL* sequences of *S. baumii* and 15 other species. (**B**) Predicted 3D structure of *Sb**PAL* showing the monomer protein (left) and homo-tetrameric protein (right). (**C**) Recombinant His-tagged *SbPAL* on the SDS-PAGE gel. (**D**) Determination of *trans*-cinnamic acid in reaction solution. The red, blue, and black peaks are standard *trans*-cinnamic acid, the reaction solution containing purified *SbPAL*, and a blank control reaction solution, respectively.

**Figure 6 jof-08-00582-f006:**
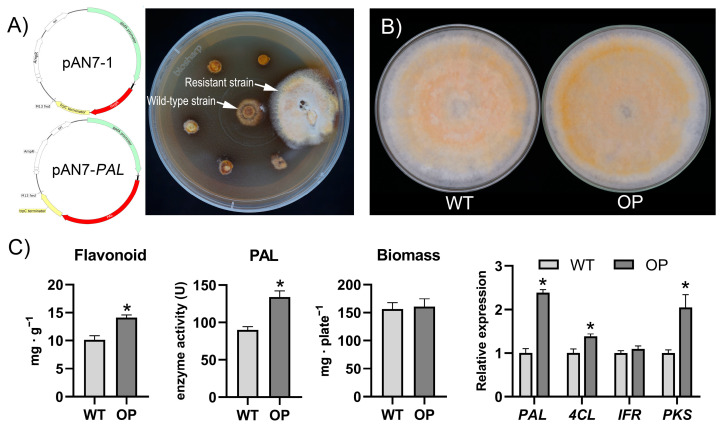
Effects of *SbPAL* overexpression on the accumulation of flavonoids in *S. baumii*. (**A**) Plasmids used for overexpression and the hygromycin-resistant colony obtained from 30 days culture on hygromycin plates. The wild-type (WT) strain is in the middle of the plate as a control. (**B**) Colonies of WT and *SbPAL*-overexpressing strain (OP). (**C**) Differences in total flavonoid content, *PAL* enzyme activity, biomass, and gene expression between the WT and OP strains. The values with * are significantly different (*p* < 0.05).

## Data Availability

All the data are mentioned in the manuscript and as Appendix A.

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
