# Peer review of "Mushrooms Do Produce Flavonoids: Metabolite Profiling and Transcriptome Analysis of Flavonoid Synthesis in the Medicinal Mushroom Sanghuangporus baumii"

_jof, 2022, doi:10.3390/jof8060582_

Round 1

Reviewer 1 Report

The aims are well defined and the reader is adequately informed about the current knowledge about the topic. Methods are very up-to-date, and well explained too. Results are clearly presented and the discussion section is based on those results supported by logical and reference-based arguments.

I believe this to be among the best research papers from the field I had read recently. Thus, I suggest it be accepted after some minor corrections.

My suggestions for correction:

1. Exclude the lines 27-34 and start from the next paragraph, since this paper does not deal with all the other compounds from mushrooms.

2. Line 87: morphological instead of morphology

3. line 105: 12 d of cultivation

4. The authors used "were" all along with  "mycelia". But, it should be "was". Please, correct it wherever it appears.

5. Also, I suggest adding a more direct title, which will undoubtedly point to the conclusion. Maybe like this: Mushrooms do produce flavonoids: Metabolite profiling and transcriptome analysis of flavonoid synthesis in the medicinal mushroom Sanghuangporus baumii

Great work!

Author Response

Thank you so much for your comments and suggestions, which are very important to us.

Here are the responses to Reviewer 1 Comments:

Point 1. Exclude the lines 27-34 and start from the next paragraph, since this paper does not deal with all the other compounds from mushrooms.

Response 1: Lines 27-34 have been deleted.

Point 2. Line 87: morphological instead of morphology

Response 2: Line 87 “morphology’’ has been changed to “morphological” .

Point 3. line 105: 12 d of cultivation

Response 3: Line 105 “12 d culture” has been changed to “12 d of cultivation” .

Point 4. The authors used "were" all along with  "mycelia". But, it should be "was". Please, correct it wherever it appears.

Response 4: Change “were” to “was” on line 100, 103, 110, 127.

Point 5. Also, I suggest adding a more direct title, which will undoubtedly point to the conclusion. Maybe like this: Mushrooms do produce flavonoids: Metabolite profiling and transcriptome analysis of flavonoid synthesis in the medicinal mushroom Sanghuangporus baumii

Response 5: The title was changed to: Mushrooms do produce flavonoids: Metabolite profiling and transcriptome analysis of flavonoid synthesis in the medicinal mushroom Sanghuangporus baumii.

Reviewer 2 Report

This manuscript written by Shixin Wang and co-authors described Metabolite Profiling and transcriptome analysis of flavonoid synthesis in the medicinal mushroom Sanghuangporus baumii. I think this paper has great novelty and the research is designed well. I advance my concern about chemical identification.

  1. For results 3.1, the authors stated 81 flavones were identified in a mushroom. However, they didn’t provide enough evidence. Figure 2 only showed two peaks corresponding to two flavones, and Table S2 showed 81 flavones, which contain a lot of isomers. The authors should provide the detail of flavone identification.
  2. Some small molecular flavones in Table S2 are non-polar compounds, I think they cannot be extracted by 70% MeOH(aq).
  3. Because the authors cannot provide adequate evidence that baumii produce so many flavones, the biosynthesis discussion (lines 350-372) seems redundant.

Author Response

Thank you so much for your comments and suggestions, which are very important to us.

Here are the responses to Reviewer 2 Comments:

Point 1.   For results 3.1, the authors stated 81 flavones were identified in a mushroom. However, they didn’t provide enough evidence. Figure 2 only showed two peaks corresponding to two flavones, and Table S2 showed 81 flavones, which contain a lot of isomers. The authors should provide the detail of flavone identification.

Response 1: Details of flavonoids identification are given in Materials and Methods 2.4, Figure 2, and Table S2.

Point 2.   Some small molecular flavones in Table S2 are non-polar compounds, I think they cannot be extracted by 70% MeOH(aq).

Response 2:  Most flavonoids are insoluble in water and soluble in organic solvents. Therefore, 70% MeOH is commonly used for metabolomics detection of flavonoid compounds. Some small molecules and non-polar flavonoids may not be well extracted with 70% MeOH, and we will consider replacing solvents for subsequent extraction experiments of these flavonoids.

Point 3.   Because the authors cannot provide adequate evidence that baumii produce so many flavones, the biosynthesis discussion (lines 350-372) seems redundant.

Response 3: It has been shown in response 1 that mushroom S. baumii can produces multiple classes of flavonoids, but more worthy of further research is how it biosynthesize flavonoids. We can only refer to the flavonoid biosynthesis pathway in plants to discuss and speculate the flavonoid biosynthesis pathway in mushrooms (lines 350-372). Hopefully this article will attract more attention to the flavonoid biosynthesis in mushrooms.